# Mitochondrial Dysfunction: A Cellular and Molecular Hub in Pathology of Metabolic Diseases and Infection

**DOI:** 10.3390/jcm12082882

**Published:** 2023-04-14

**Authors:** Tapan Behl, Rashita Makkar, Md. Khalid Anwer, Rym Hassani, Gulrana Khuwaja, Asaad Khalid, Syam Mohan, Hassan A. Alhazmi, Monika Sachdeva, Mahesh Rachamalla

**Affiliations:** 1School of Health Sciences and Technology, University of Petroleum and Energy Studies, Bidholi, Dehradun 248007, India; 2Chitkara College of Pharmacy, Chitkara University, Rajpura 140401, India; 3Department of Pharmaceutics, College of Pharmacy, Prince Stattam Bin Abdulaziz University, Al-kharj 11942, Saudi Arabia; 4Department of Mathematics, University College AlDarb, Jazan University, Jazan 45142, Saudi Arabia; 5Department of Pharmaceutical Chemistry and Pharmacognosy, College of Pharmacy, Jazan University, Jazan 45142, Saudi Arabia; 6Substance Abuse and Toxicology Research Centre, Jazan University, Jazan 45142, Saudi Arabia; 7Medicinal and Aromatic Plants and Traditional Medicine Research Institute, National Center for Research, P.O. Box 2404, Khartoum 11123, Sudan; 8Center for Transdisciplinary Research, Department of Pharmacology, Saveetha Dental College, Saveetha Institute of Medical and Technical Science, Saveetha University, Chennai 602105, India; 9Fatimah College of Health Sciences, Al Ain P.O. Box 24162, United Arab Emirates; 10Department of Biology, University of Saskatchewan, 112 Science Place, Saskatoon, SK S7N 5E2, Canada

**Keywords:** mitochondria dysfunction, cancer, diabetes mellitus, obesity, metabolic disorders, infection

## Abstract

Mitochondria are semiautonomous doubly membraned intracellular components of cells. The organelle comprises of an external membrane, followed by coiled structures within the membrane called cristae, which are further surrounded by the matrix spaces followed by the space between the external and internal membrane of the organelle. A typical eukaryotic cell contains thousands of mitochondria within it, which make up 25% of the cytoplasm present in the cell. The organelle acts as a common point for the metabolism of glucose, lipids, and glutamine. Mitochondria chiefly regulate oxidative phosphorylation-mediated aerobic respiration and the TCA cycle and generate energy in the form of ATP to fulfil the cellular energy needs. The organelle possesses a unique supercoiled doubly stranded mitochondrial DNA (mtDNA) which encodes several proteins, including rRNA and tRNA crucial for the transport of electrons, oxidative phosphorylation, and initiating genetic repair processors. Defects in the components of mitochondria act as the principal factor for several chronic cellular diseases. The dysfunction of mitochondria can cause a malfunction in the TCA cycle and cause the leakage of the electron respiratory chain, leading to an increase in reactive oxygen species and the signaling of aberrant oncogenic and tumor suppressor proteins, which further alter the pathways involved in metabolism, disrupt redox balance, and induce endurance towards apoptosis and several treatments which play a major role in developing several chronic metabolic conditions. The current review presents the knowledge on the aspects of mitochondrial dysfunction and its role in cancer, diabetes mellitus, infections, and obesity.

## 1. Introduction

Mitochondria, also known as the power-generating organelle of the cell, are highly responsible for metabolizing organic molecules and releasing energy for cellular activities in the form of adenosine triphosphate (ATP) [1]. In mammals, most of the redox reactions that take place for synthesizing ATP within mitochondria are in the form of nicotinamide adenine dinucleotide (NADH), flavin adenine dinucleotide (FADH2), and tricarboxylic acid (TCA) cycle-mediated decrease in the generation of coenzymes due to the acceptance of electrons upon breakdown of organic substances [2]. The redox energy stored as NADH and FADH2 within the mitochondria is changed into chemical energy in the form of ATP by the action of four protein complexes, which makes the electron transport chain (ETC) [3]. The complexes I and II receive electrons donated from FADH2 and NADH and further oppress complexes III and IV, finally donating oxygen molecules and releasing water. The shuffling of electrons through complexes I, III, and IV with the help of electron carriers produces the movement of protons from the inner matrix of mitochondria to the intermembrane spaces, which produces a positive charge in the mitochondrial intermembrane space and negative charge in the matrix of mitochondria. This electrochemical difference mainly drives the production of ATP within the mitochondria [4]. The current article presents a recent comprehensive review of the role of mitochondrial dysfunction in the onset of several chronic life-altering disorders such as cancer, diabetes mellitus, obesity, and infections. This article explains in brief the life cycle of mitochondria, including their processes of fusion, fission, and autophagy. This article also highlights the role of mitochondria in the metabolic processes of the body and concisely describes their validation and co-occurrence with nuclear DNA. An extensive study of the recent literature was performed in the preparation of this article to present the readers with the various roles of mitochondria in the management of several chronic disorders elaborately discussed in a single review paper, which brings novelty to this article.

### 1.1. The Mitochondrial Life Cycle: Fusion, Fission, and Autophagy

The pre-existing mitochondrion increases in volume, elongates in size, and eventually forms two physically separate mitochondria organelles and therefore cannot be synthesized de novo [5]. This process of mitochondria formation is known as mitochondrial fission. The organelle also possesses the ability to fuse by joining membranes and compartments to form a single bigger mitochondrion [6]. The process of mitochondrial fission and mitochondrial fusion is a lifetime cycle, and the organelle stays in a dynamic flux between the two processes at any given time point. The balance between the two processes leads to diversity in the size and structure of mitochondria which can be observed visibly while assessing the morphology of the organelle [7]. Several GTPase proteins, namely Mfn1, Mfn2, OPA1, are also involved in the process of mitochondrial fusion. Mfn1 and Mfn2 are known as mitofusins. The mitofusin proteins are located on the external membrane of mitochondria and facilitate the fusion of the outer membrane, whereas the OPA1 is mainly localized in the internal membrane and facilitates the fusion of the inner membrane of the mitochondria. For the occurrence of fusion, both Mfn1 and Mfn2 proteins must be present on the membranes of both the mitochondria, whereas the presence of OPA1 on the inner membrane of any one of the mitochondria is sufficient for internal membrane fusion [8]. The inner and outer membranes of the mitochondria fuse concurrently while they come together, preventing the mixing of the intermembrane space with the contents of the matrix and hence preserving the functioning of the organelle [9]. Dynamin-related protein (Drp1), a large GTPase, has been found to play an implicit role in the process of mitochondrial fission by catalyzing membrane pinching and produces an action similar to vesicle endocytosis and peroxisomal fission. The Drp1 protein is confined within the cytosol, prompting the expression of a receptor bound to the mitochondrial membrane for the recruitment of the protein. Mitochondrial fission proteins have been found to evidently play a significant role as Drp1 receptors and induce their further recruitment. The process of mitochondrial fission remains a hot topic of research because the involvement of proteins in the processes and their further interaction at mitochondria–endoplasmic reticulum contact sites is still to be elucidated. The mutation and deficiency of proteins involved in the fusion and fission of mitochondria can play an eminent role in the pathophysiology of several chronic diseases [10]. The process of mitochondrial fission crucially maintains mitochondrial respiration, as apart from its role in division of mitochondria, it is also employed in the waste disposal system of the mitochondria. A mitochondrion during its lifespan can store debris and experience damage in the form of excessive reactive oxygen species and dysfunctional proteins, which causes the uncoupling of the electron transport chain. The molecular damages and debris within mitochondria can be discarded by segregation through the mitochondrial fission process followed by its employment in autophagosome for mitophagy [11]. The mechanism for mitophagy in mammals is believed to be a vital pathway for the prevention of metabolic disorders as defects in mitophagy have been found to be linked to human pathophysiology, and it is also associated with DNA repair, age, tissue injury, and metabolic processes. Autophagy plays a significant role in the functioning of mitochondria. The average half-life of mitochondria in the mammalian brain is approximately 10 to 25 days. The turnover of mitochondria during the condition of starvation can be accelerated via autophagic processes explicitly known as mitophagy. The depolarization of the membrane of mitochondria presents autophagy induction by depriving the hepatocytes of nutrients and the further co-localization of depolarized mitochondria with autophagosomes. For the maintenance of mitochondrial homeostasis, the remodeling of mitochondria via mitochondrial fission, mitochondrial fusion, and mitophagy is crucial. Mitophagy aids in the clearance of damaged mitochondrial debris by attenuating necrosis or apoptosis and further preventing the involuntary release of apoptosis inducing factor (AIF), cytochrome C, and other apoptotic factors. Mitophagy is mainly regulated either in combination with general macro autophagy or specifically through selective mitophagy genes. The decreased concentration of intracellular ATP activates AMPK, which further causes the phosphorylation of two Atg1 homologs, namely ULK1 and ULK2. These activated phosphorylates further stimulate the processes of macro autophagy and mitophagy. The hepatocytes deficient in ULK1 or AMPK present the accretion of p62, aggregated ubiquitin, and abnormal mitochondrial cells. Several proteins, such as PINK1, Bcl-2 adenovirus E18 19-kDa interacting protein (BNIP), parkin, and Atg32, interact with mitochondrial fusion and fission processes and selectively regulate mitophagy. Atg32, a yeast mitophagy definite protein, is confined to the outer membrane of mitochondria where it positively interacts with Atg11, which further interrelates with Atg8 and combines mitochondria to the autophagosome. The autophagy induced mainly by BNIP proteins prevents cell death as it is independent of uncoupling proteins or the ratio of mitochondrial DNA or nuclear DNA. Hence, an evident role of autophagy and mitophagy has been observed in the functioning of mitochondria. Therefore, a suppression in mutations of mitochondrial DNA can be achieved by the process of mitophagy. A brief description of mitochondrial life cycle has been represented in Figure 1.

### 1.2. Role of mtDNA in Maintenance of Metabolic Function

Mitochondria have existed within host mammalian cells for more than two billion years, and the integration of elements of mtDNA into the nuclear genome seems to be a continuous process. The organelle comprises of approximately 2 to 10 mtDNA on an average, depending on their position in the mitochondrial fission–fusion cycle. Through studies, it has been discovered that mutation-suppressing components such as intron sequences have been lacking in the human mtDNA, resulting in ten times higher chances of mutations. The incidences of primary mutations also increase abundantly in the sequences of mtDNA due to its small-sized molecules, which further results in a deficiency of the ETC and impairs the functioning of mitochondria [12]. Hence, in lieu of a paramount role of mtDNA in the regulation of metabolic processes, it is essential to maintain mitochondrial health for the prevention of several diseases. The reactive oxygen species (ROS) generated within the organelle pose a significant impact on mutations and damage the mtDNA. The power-generating organelle does not possess any mechanisms of its own to protect its DNA; however, it comprises several nuclear-encoded proteins which aid in the repair and rejuvenation of the mtDNA [13]. The finest procedure characterized for the restoration of mtDNA is the excision of base pairs in which the damaged or improperly paired base is expunged accompanied with further filling of the gap and ligation of a new base to the addressed nucleotide. Upon mitochondrial dysfunction, the nuclear proteins involved in the excision and repair of mtDNA localize, indicating their existence within the organelle [14]. The reactive oxygen species-mediated sugar damage further causes breakage in the single and double strands of the mtDNA, releases torsional stress due to failure of topoisomerases, and impacts transcription. The inability of mitochondria to repair their damaged DNA leads to the segregation of the damaged mtDNA into a region which instantly reduces the respiratory activity upon fission and is targeted for mitophagy. In a rare scenario, excessive mutation and an inability to repair DNA can trigger apoptosis of the entire cell, indicating a significant role of the maintenance of mtDNA in growth and mitochondrial fitness [15].

### 1.3. Nuclear DNA and Mitochondrial DNA Cooccurrence

Equating the DNA of mitochondria to nuclear DNA is crucial while selecting potential donors for the treatment of several health problems in later life. The DNA which makes the body’s blueprint is confined within our cells and is called nuclear DNA. Nuclear DNA mainly presents all the characteristics exclusive to us as individuals and encodes all the proteins which are required for the physiological processes in our body [16]. Every cell contains energy-generating organelles called mitochondria, which are the chief powerhouse source of the cell. The mitochondrial DNA codes a tiny amount of the mitochondria, makes up 0.1% the entire genome of humans and is passed on through generations [17]. The mutation rate of the mitochondrial DNA is much higher than that of the nuclear DNA, and mutations within the genome of mitochondria occur frequently. The nuclear DNA shape alterations in the mitochondrial DNA signify a subtle relationship between the nucleus and the mitochondria of cells. The replacement of mitochondria is emerging as a new treatment which enables mothers to prevent severe metabolic complications which arise due to mutations in mitochondrial DNA in their children. This new treatment includes matching the nuclear genome with the mitochondrial genome of the individual donor, a process which is similar to organ transplantation. The nuclear and mitochondrial genome are also linked with each other in several ways, such as genetic transfer, regulation of genetic expression, and encoding together the composite proteins, meaning that the proteins are encoded within the mitochondria and are a part of the nucleus. An example that comprises both nuclear- and mitochondrial-encoded subunit proteins is cytochrome oxidase [18]. The mitochondrial genome is categorized into three types: transfer RNA (tRNA), ribosomal RNA (rRNA), and protein-encoding genes. Over 3–67 protein genes and 27 tRNA genes are encoded by mitochondrial DNA. Genetic loss and diversity in a number of encoded genes in mitochondria suggest the role of nuclear-encoded proteins and mtDNA transferred into the genome of the nucleus. The characterized mitochondrial genomes comprise two chief ribosomal RNA genes which encode all the small and large subunits of the ribosome. The extent of coding of tRNA by mtDNA fluctuates, and it was found that a few tRNA genes are not encoded by mitochondria and are needed for the synthesis of proteins. The mitochondria utilize nuclear-encoded tRNA for the mitochondria-mediated synthesis of proteins, indicating the co-occurrence of nuclear and mitochondrial genomes in metabolic functions of the body.

### 1.4. Functional Validation of Mitochondria

The disruption of the functioning of mitochondria is a clinical feature of early-stage programmed cell death. This disturbance of mitochondria comprises altered membrane potential and redox potential, which are the key features of healthy mitochondria. The internal membrane potential of mitochondria essentially uptakes and stores calcium, regulates generation of ROS, is involved in detoxification, and is involved in ATP synthesis by oxidative phosphorylation. Hence, the depolarization of the mitochondrial membrane is a great indicator of its dysfunction and is chiefly associated with the toxicity of drugs. Alterations in the potential of the membrane, including decreased ATP/ADP ratio, increased matrix levels of calcium, oxidative stress, and secretion of cytochrome c into the cytosol, have also been stated to be linked with the transitioned permeability of mitochondria leading to ion and small molecular-mediated disrupted homeostasis. The altered functioning of mitochondria can be observed with the use of fluorescence-based assays, including the quantification of mitochondrial calcium, superoxide, and mitochondrial potential transition. Multiparametric assays have always been found to be beneficial in identifying markers specific to caspase proteases. The assays can be performed by screening membrane potential and metabolism, production of superoxide, calcium, and the transitioned permeability of mitochondria with the help of different reagents, namely JC-1 dye, Rhod-2 AM reagent, MitoSox red reagent, MitoSox green reagent, Mito tracker flow cytometry, and Mito tracker imaging [19,20].

## 2. Pathophysiological Role of Mitochondrial Dysfunction in Cancer

Cancer is a multifaceted disease which involves several alterations in the genomic, epigenomic, proteomic, transcriptomic, and metabolic levels. Otto Warburg in the year 1924 first described that cancer cells comprise reformed metabolism and metabolize glucose anaerobically even in the presence of oxygen, with a subsequent increase in the production of lactate for growth and survival [18]. The alteration of metabolism is marked as the chief hallmark of tumorigenesis and regulates growth, migration, and invasion. The remodeling of the metabolic profile is crucial for the existence of tumor cells in an unwelcoming environment with curbed nutrients, low levels of oxygen, and immunological surveillance, which further enhances the proliferation of tumor cells and accelerates their biological activity. Besides the requirement for energy to sustain their replication, the cancer cells also boost neosynthesized macromolecules to maintain a redox balance and improve their capabilities. Several hypotheses state that tumorsnot only involve glucose-dependent metabolism but also benefit from alternate oxidized substrates such as serine, glutamine, fatty acids, etc. The cells located in the microenvironment of tumor growth can influence the behavior of cancer cells and provide energy, thereby impacting the metabolic demand of cancer cells. Hence, to encourage the growth of the tumor, it is essential to have functionally active mitochondria, mostly due to their biosynthetic activity and not limited to pro-energetic features. Mitochondria, at several stages, satisfy the metabolic needs of cancer cells, significantly modify the synthesis of bioenergetics such as NADPH and ATP, and convert the various available nutrients into central building blocks which are required for the growth and functioning of the cells [19,21]. Hence, mitochondria within cancer cells can be targeted for the management of this debilitating disease and can be highlighted as its future treatment [20,22]. 

### 2.1. Mitochondrial Reactive Oxygen Species and Cancer Metabolism

Mitochondria are the basis for the production of intracellular ROS because about 1 to 2% of the molecular oxygen which is utilized in the process of oxidative phosphorylation metabolizes into anion superoxide. Several mitochondria-linked enzymes, such as pyruvate dehydrogenase (PDH) [23], acyl-coA dehydrogenase [24], α-ketoglutarate dehydrogenase [25], and glycerol 3 phosphate dehydrogenase [26], also stimulate the production of ROS. The levels of mitochondrial ROS in healthy cells are controlled and exhibit a pivotal role in various cellular processes incorporating autophagy, differentiation, and metabolic acclimatization. The activation of oncogenes, hypoxia, loss of tumor suppressor proteins, and cancer-producing mutations in the TCA cycle lead to the abnormal synthesis of mitochondrial ROS, which give further retrograde signals and sustain the life of cancer cells. The mitochondrial ROS shape the entire process of oncogenesis, starting from the initiation of the tumor, followed by its proliferation and metastasis due to the accretion of mutations in mitochondrial or nuclear DNA which directly impact multiple biological processes including resistance to apoptosis and reprogramming of metabolic functions. Moreover, the mitochondrial ROS have a tendency to stimulate different potential oncogenic signal pathways such as the Akt/NF-κB mitochondrial transcription factor B2-mediated signaling pathway and epidermal growth factor receptor signaling pathway, which cause the cascade signaling of different trigger factors, proliferation of cancer cells, and are also associated with metastatic dissemination [27]. However, through several studies, it was established that metastatic dissemination can be limited by oxidative stress in melanoma and lung cancers. Tumor cells prevent the accumulation of ROS by expressing high levels of antioxidant proteins and push cancer cells in the direction of a proliferative state by simultaneously dodging ROS-mediated mitochondrial permeability transition (MPT)-associated cellular death. A healthy balance between the generation and scavenging of mitochondrial ROS permits cancer cells to stay in their tumorigenic state with the desired levels of ROS [28]. A transcriptional factor, namely nuclear factor 2 (derived from erythroid), is the master genetic regulator which mitigates oxidative stress by binding to antioxidant response elements present in the gene promoters. The exposure to mitochondrial ROS provokes the rapid stimulation of the Nrf2 protein upon the deterioration of its allosteric inhibitor, namely kelch-like ECH associated protein 1 (KEAP1). Nrf2 was initially believed to be a tumor suppressor; however, following recent studies, it has been shown to possess pro-tumoral activity which not only convenes resistance towards oxidative stress but also regulates the production of ROS via NADPH oxidase and directly sets off metabolic processes associated with cancer. The production of mitochondrial ROS due to hypoxic conditions leads to the activation of hypoxia-inducible factor 1 (HIF1). HIF1 advances to metabolic swing from oxidative phosphorylation to glycolysis by enhancing the manifestation of glycolytic enzymes and expediting metastasis and tumorigenesis. The connection between mitochondrial ROS and HIF1 is highly complex as the over-synthesis of mitochondrial ROS activates HIF1, whereas the oxidative stress is alleviated upon activation of the glycolytic program in a reparation manner. In various forms of cancer, a decrease in the production of mitochondrial ROS by HIF1 promotes the growth of tumors and facilitates the existence of metastatic cells [29]. Overall, a functional role of mitochondrial ROS can be indicated depending upon the stage and type of cancer, and it can be suggested that mitochondrial ROS can preferentially initiate the signaling of various pro-tumoral proteins and lead to metabolic reprogramming. Hence, the targeting of mitochondrial ROS and antioxidants can be presented as a beneficial and novel therapy to manage cancer. 

The oxidative stress induced by mitochondria is inversely related to longevity in animal models, and defective bioenergetics in mitochondria have been implied in various chronic diseases. The association between ATPase inhibitory factor 1 (IF1) and H+-ATP synthase is dependent upon the mitochondrial matrix. The IF1 in a low pH environment binds to α, β, and γ subunits of the H+-ATP synthase, blocks the hydrolysis of ATP, and prevents the unusable wastage of energy. Substituting histidine 49 with lysine in the sequence of IF1 produces a mutated IF1 (H49K) which is similar in actively inhibiting ATP hydrolase but is less sensitive to pH. The production of an antiparallel α-helical coil in the C-terminal area stimulates and dimerizes IF1 and places the monomeric inhibitory regions N-terminus on the dimeric opposite ends, which permits the binding of the two domains of IF1–ATP synthase simultaneously and inversely monitors the ROS production mediated by mitochondria. 

### 2.2. Mitochondrial Metabolism Deregulation-Mediated Generation of Tumor-Related Proteins and Oncometabolites

The accumulation of metabolites in response to catabolic and anabolic process dysfunction leads to alterations in mitochondria, which is a sign of pathophysiological modifications and involves abnormal signaling and the establishment of disease phenotype. The deregulation of the metabolism of mitochondria can arise not only from mutations in somatic mitochondrial DNA but also through defects in the encoding of nuclear mitochondrial enzymes. The mitochondrial enzymes whose mutations are reflected as pro-tumorigenic in several types of cancer are either a component of oxidative phosphorylation or the TCA cycle and also play a significant role in pathways of biosynthesis [30]. The respiratory complex II, also named succinate hydrogenase (SDH), localized in the inner mitochondria membrane, aids in the transformation of succinate to form fumarate. In the setting of cancer, SDH is affected by loss of function (SDH LOF) mutations, which stimulate succinate accumulation largely in the mitochondria and subsequent leaks within the cytoplasm [31]. Recently, the overexpression of the enzyme has been found in patients with prostate cancer. Undeniably, fumarate produced upon the oxidation of succinate can withstand the impairment in the metabolism of mitochondria via the rewiring of oxidative phosphorylation, supporting the TCA cycle and stimulating the production of ATP. However, the mechanism responsible for the activity is still to be elucidated. The mutations causing SDH LOF have been associated with gastrointestinal stromal tumors, breast cancer, neuroblastoma, thyroid tumors, renal carcinoma, and pheochromocytoma. Excessive levels of succinate inhibit prolyl hydroxylase (PHD), an oxygen dependent enzyme that controls the expression of HIF1 based on the homeostasis of oxygen and apparently stabilizes the initiation of HIF1 under normal conditions [32,33]. Defects in SDH can produce a state of pro-oncogenic pseudohypoxia, and the accretion of succinate can alter the epigenetic expression by inhibiting α-ketoglutarate (α-KG) and DNA demethylation, thereby restoring the proliferation of cancerous cells [34]. 

Another nuclear-encoded enzyme of the TCA cycle, namely fumarate hydratase (FH), confined within the mitochondrial matrix, converts fumarate to malate. The SDH mutation decreases the respiratory rate in response to the loss of functioning of the TCA cycle, and increases the production of lactate, hence reinforcing the glycolytic metabolic switch [35,36]. The enzyme commonly mutates in the hereditary environments of leiomyomatosis renal cell carcinoma. The oncogenic action of FH, just like SDH, mainly depends upon fumarate accumulation, and leads to HIF-1α stabilization upon restraining the activity of PHD, which mediates epigenetic mutations in response to the α-KG-mediated inhibition of histone and DNA methylation. Both FH and SDH can be regarded as tumor suppressors, as the deposition of succinate and fumarate promotes the proliferation of cells, usually directed towards similar targets. Nonetheless, these oncometabolites mutate and exercise definite functions in several types of cancer and can be employed as a successful therapy for disease management. Several more enzymes in the TCA cycle, such as succinate dehydrogenase [37], fumarate hydratase [38], isocitrate dehydrogenase [39], SIRT3 [40], etc., also present alterations which directly support the proliferation and survival of tumor cells and are being explored for their anti-cancerous effects [41]. 

### 2.3. Mitochondria as an Upcoming Target for Management of Cancer

The transposition of various processes in cancer cells exposes several vulnerable mechanisms which can be targeted to produce therapeutic strategies. However, the heterogeneity of tumors, despite the presence of several pathways, is a limitation to the progress of these therapeutic interventions [42,43]. Metformin, taking bioenergetics into account, possesses an ability to inhibit complex I of the ETC in mitochondria, which further represses the production of ATP and disrupts the balance of NAD+/NADH and reduces the consumption of oxygen. The process further induces the activation of AMPK in response to the diminished activity of TCA and further leads to autophagy. The tumor cells can compensate for several stress situations through a variety of mechanisms, including an increased uptake of glucose and glycolysis and further switching to the utilization of glutamine [44]. Hence, to overcome this effect, a combination of glycolytic inhibitor 2 deoxyglucose (2-DG) with metformin was recommended for the management of cancer. The synergistic combination successfully decreased the levels of ATP and further channeled the activation of proliferative signaling processes, thereby suppressing the adverse effects associated with the intake of a high-dose treatment with single drugs. The therapeutic potential of 2-DG can be enhanced by combining it with effective radiotherapy and chemotherapy for better management of cancer. The glucose analogue 2-DG competes with circulating glucose to bind to hexokinases and elevates oxidative stress and apoptosis, induces autophagy, and ultimately reduces the growth of cancer cells [45,46]. The ability of 2-DG to enhance autophagy mainly acts as a limiting factor for its efficacy as an anti-cancer agent because it sustains the survival of cancer cells, making it crucial to simultaneously target several mitochondrial pathways. Several new complex I inhibitors, namely IACS-010759, BAY87-2243, and mitochondrially targeted tamoxifen (Mito Tam), have also been proposed to promote the death of cancer cells and reduce their multiplication. Vitamin E succinate (VES) is another mitochondrion-targeting modified drug, which inhibits both complex I and complex II, promotes the production of ROS, induces apoptosis by restoring Fas (CD95), transforms growth factor beta-mediated signaling pathways, further activates the c-Jun N-terminal kinase pathway, and is significantly being used for the management of breast cancer [47]. Lonidamine, another complex II inhibitor, amends the TCA cycle, regulates the metabolism of glutamine in cell lines of melanoma, and is employed in combination with other chemotherapy drugs, thereby enhancing their overall efficacy and response towards cancer management [48]. 

Further investigations are ongoing to understand the mechanism associated with the regulation of mitochondria in altering the metabolism of cancer cells to build successful promising therapies for the management of this debilitating disorder (Figure 2 and Figure 3).

## 3. Pathophysiological Role of Mitochondrial Dysfunction in Diabetes Mellitus

Hyperglycemic conditions and type 2 diabetes are directly related to an increase in oxidative stress. An excessive production of reactive oxygen species and their further subsequent alterations in the redox state and cellular homeostasis is a critical presentation of type 2 diabetes. Mitochondria are the major site of the production of ATP and act as a chief source of reactive oxygen species. The organelle increases the production of reactive oxygen species when the levels of glucose are high and increases oxidative stress, which leads to tissue damage. Mitochondrial impairment with age can cause insulin resistance in the body in response to altered fatty acid oxidation and impaired glucose, protein, and lipid metabolism, which act as hallmarks for the onset of the disease [49]. The biogenesis of mitochondria modulates the energy balance and increases the production of reactive oxygen species by the electron transport chain in response to conditions of hyperglycemia, which further aggravates the pathological conditions and induces the associated complications, such as nephropathy, neuropathy, and retinopathy, and microvascular complications such as myocardial ischemia and stroke [50]. The mitochondria have been found to be major regulators of the secretion of insulin, and mutations in mtDNA are directly involved in the development of type 2 diabetes mellitus. ATP production significantly decreases upon mutations, which leads to the impairment of biochemical characteristics and further alters the synthesis of mitochondrial ATP synthase [51]. Insulin is mainly synthesized by islets of β-cells in the pancreas based on the circulating levels of glucose in the body. Higher glucose concentrations produce oxidative phosphorylation in the β-cells, which further increases the ratio of ATP/ADP and prevents the depolarization of the plasma membrane by inhibiting potassium channels and stimulating the release of calcium ions within the cells, which triggers the secretion of insulin in the bloodstream. Therefore, the role of mitochondria in modulating the secretion of insulin has evidently been documented [52]. 

### 3.1. Insulin Resistance and Mitochondrial Dysfunction

The development of resistance to insulin is a common feature of type 2 diabetes mellitus and can also be attributed to the dysfunction of mitochondria. An increase in the peroxidation of lipids and the inhibition of function has been determined in the skeletal muscle of diabetic patients. Moreover, the suppression of genes involved in the biogenesis of mitochondria and oxidative phosphorylation has also been found in individuals with type 2 diabetes mellitus. Recently, the suppression of the oxidative phosphorylation pathways of mitochondria has also been documented globally in the obese population, which is simultaneously linked to the concurrent downregulation of mtDNA, dependent protein translation, and oxidative phosphorylation mechanisms [53].The repression of the oxidation of fatty acids, the production and breakdown of ketone bodies, and the TCA cycle were observed to be inversely linked to insulin resistance, adiposity, and an increase in inflammatory cytokines [54]. Insulin sensitivity can be improved by regular exercise, mitochondrial-targeted antioxidants, and counted caloric diet consumption, which can further manage mitochondrial functioning and improve type 2 diabetes. The commonly used antidiabetic drugs metformin and resveratrol have also been found to protect the integrity of the mitochondria by inhibiting the activity of the DRP1 enzyme and preventing the activation of NLR family pyrin domain containing 3 (NLRP3) inflammasomes (mediating activation of caspase-1 and proinflammatory cytokine secretion) via the suppression of endoplasmic reticulum-originated stress and portrayed beneficial effects by protecting the cellular functions under the state of hyperglycemia [55]. Several mitochondrial-targeted antioxidants such as SS-31 have also displayed beneficial effects which successfully modulate the mitochondrial membrane potential and alterations in ATP. It further inhibits the expression of NADPH oxidase-4 (Nox 4) and TGF-β1 and prevents the activation of p38 mitogen-activated protein kinase (p38 MAPK) and NADPH oxidase activity in the mesangial cells under hyperglycemic conditions. Several studies have also indicated that the functioning of mitochondria and sensitivity towards insulin can be significantly improved by regular exercise and consuming a calorie deficit diet [56]. The changes in mitochondrial membrane potential are related to the induction of oxidative stress in the β cells of the pancreas. Alterations in β cells induce changes in mitochondrial dynamics, impair the secretion of insulin mediated by glucose, and lead to the onset of type 2 diabetes. The processes of mitochondrial fusion and mitochondrial fission are continuously reformed within the β cells of patients with type 2 diabetes, as hyperglycemia and increased concentration of palmitate inhibit the mitochondrial oxygen consumption and lower mitochondrial fusion. PTEN-induced kinase 1, a mitochondrial serine/threonine protein encoded by the PINK1 gene, alleviated insulin resistance by lowering the production of ROS mediated by the MAPK pathways. Diabetes is also linked to the condensed appearance of mitofusin-2 proteins, which are closely associated with the functioning of the power-generating organelle in skeletal muscles. Mitochondrial fusion was decreased and mitochondrial fission was increased in the leukocytes of patients with type 2 diabetes, and both processes were stressed in patients with inadequate glycemic control [57]. Leukocyte amplification, similarly to HbA1c values, affects the dynamics of mitochondria by inducing leukocyte–endothelial interactions in diabetic patients, leading to lowered mitochondrial fusion and intensified mitochondrial fission. It has been hypothesized that inadequate glycemic control during diabetes modifies the mitochondrial dynamics, ultimately fosters the interaction between leukocyte and endothelial cells, and could be the origin of cardiovascular diseases [58]. The endoplasmic reticulum membranes coupled to mitochondria reorganize under the conditions of insulin dependence and reduce the consumption of oxygen, thereby elevating oxidative stress. 

A significant change in the size and shape of mitochondria has been observed in patients with diabetes, as it was found that the organelle was of a smaller size in type 2 diabetic patients compared to healthy individuals. Hyperglycemic states also induce fragmentation of mitochondria in the cells of organs such as the liver, heart, pancreas, etc. The mitochondria were also found to be swollen and disrupted in the hepatocytes of patients with developed insulin resistance [59]. Elevated levels of glucose induce the production of ROS and cause mitochondrial fragmentation, which can be prevented by decreasing the circulating levels of ROS and recovering the activity of mitochondria. During the conditions of very high levels of glucose, significant upstreaming of alterations in the morphology of mitochondria occurs, which is the main contributing factor in the synthesis of ROS and emphasizes the major role of mitochondrial dynamics in regulating mitochondrial activity. It should be noted that abnormality in the homeostasis of glucose must not always be considered the cause, but rather be looked upon as a consequence that leads to disturbed mitochondrial dynamics. Type 2 diabetes is linked to the fragmentation of the mitochondrial network in the myocardium and decreases the expression of the mitofusin-1 gene [60]. Therefore, it has been established that the secretion of insulin upon stimulation by glucose and the mitochondrial dynamics are clearly adapted to generate precise levels of expression of the fission proteins.

### 3.2. Targeting of Mitochondrial Processes of Mitochondrial Fission and Mitochondrial Fusion to Prevent Mitochondrial Dysfunction

Over time, an increased interest has been developed in synthesizing compounds directed towards the mitochondrial processes of fusion, fission, and mitophagy, highlighting the significance of these processes in the prevention of several diseases. For instance, mitochondrial division inhibitor-1 (mdivi-1), inhibitor of GTPase activity of DRP1, is being evaluated as a beneficial compound and potent inhibitor of mitochondrial fission alongside dynasore, P110, and S3. Mdivi-1 inhibits apoptosis (where dynasore Mdivi-1 and P110 are dynamin inhibitors, whereas S3 causes Mfn deubiquitination to foster mitochondrial fusion) by obstructing the release of cytochrome c in HeLa cells and ameliorates renal injuries and myocardial infarction under the conditions of ischemia. The impairment in mitochondria and the production of ROS by palmitic acid can also be reversed by mdivi-1. Mdivi-1 improves sensitivity towards insulin and prevents the onset of insulin resistance [61,62]. It reduces the fragmentation of mitochondria, oxidative stress, atherosclerosis, and inflammation and improves the endothelial functioning in mice induced with diabetes mellitus, suggesting that it could be a great therapeutic approach to managing the macrovascular complications associated with the disease. 

Long term treatment with mdivi-1, in scenarios where it is used for more than 24 h, has demonstrated its ability to inhibit the functioning of mitochondria and reduce mitochondrial mass, induce apoptosis, and lower vascular cell bioenergetics and the proliferation of cells during myogenic differentiation. Mdivi-1, in response to its cell division inhibitory action, also provokes the death of cells in cancer cells, indicating its potential as an anti-cancer compound [63]. Dynasore is another small-sized molecular compound, which disrupts the activity of GTPase of dynamin and further obstructs the endocytic pathways which depend upon it. Dynasore can also inhibit mitochondrial fission because DRP 1 is a dynamin-like protein and can prevent oxidative stress and further enhance the survival of cardiomyocytes under the conditions of ischemia and reperfusion. It maintains the morphology and functioning of mitochondria, stores ATP concentrations even under conditions of oxidative stress, and protects against cardiac lusitrophy and cell damage. The cytosolic dynamin-related protein-1 (DRP1) requires connectors such as mitochondrial fission factor (MFF) and mitochondrial fission 1 protein (FIS1) to harbor mitochondria for mitochondrial fission. Hence, it becomes important to cultivate molecules that successfully inhibit this interaction. P110 is another small peptide inhibitor which eases the activity of the DRP1 enzyme and blocks the interactions between DRP1 and FIS1 enzymes in the neurons, thereby presenting beneficial effects in mitochondrial functioning and morphology. The peptide molecule has also been found to reduce the loss of neurons and demonstrated its neuroprotective effects in the management of Parkinson’s disease. It has also presented beneficial effects in animal models of brain damage induced by reperfusion/ischemia by preventing the interaction between p53 and DRP1 enzymes, ultimately diminishing mitochondria-mediated necrosis and brain infarction. Another compound, namely 15-oxospiramilactone (S3), a diterpenoid derivative, has been found to be beneficial for mitochondrial dynamics, acts as a potent anticancer molecule, and inhibits signaling of Wnt/β-catenin and induces BAX/BAK-independent apoptosis [64,65]. The molecule also targets deubiquitinase USP30, an iso-peptidase in mitochondria, which controls the mitochondrial morphology in response to the action of MFN1/2. S3 enables the restoration of functioning of mitochondria by inducing fusion in MFN1 and MFN2 knockout cells, indicating its chief role in the management of diseases related to insulin resistance and emerging as a potent therapeutic approach. 

The exploration of several pathways and approaches has led to the development of targets which can successfully be used in the management of diabetes mellitus through alterations of mitochondrial metabolic and biogenesis processes [66] (Figure 4).

## 4. Pathophysiological Role of Mitochondrial Dysfunction in Obesity

During the differentiation of adipocytes, the biogenesis of mitochondria increases dramatically, suggesting its implicated role in the induction of obesity. Moreover, defects in the oxidation of fatty acids, adipokine secretion, and the altered homeostasis of glucose have also been encountered due to the dysfunction of mitochondria in mature adipocytes [67,68,69]. The oxidative capacity of brown adipocytes decreases in diet-induced obese patients, which causes the impairment of thermogenesis [70]. Several mitochondrial enzymes have been found to be essential for the proficient metabolism of lipids because the organelle acts as the chief site for the oxidation of fatty acids. A negative balance in energy classically enhances lipolysis in the white adipose tissues (WAT), thereby offering nonesterified fatty acids (NEFA) as a substrate for fatty acid oxidation (FAO), accompanied with a sensitization towards insulin in the liver and skeletal muscles [71]. Excessive nutrition for prolonged periods results in the accumulation of NEFA, dysfunction in the mitochondria, and resistance towards insulin. Considering the significant role of mitochondria, disorders in the organelle can disturb the fat-storing ability of the body and produce multiple symmetrical lipomatosis [72]. The accumulation of triglycerides is increased, and the FAO-mediated uptake of glucose in pre-adipocytes is decreased when mitochondrial respiration is inhibited. The uncoupling of mitochondria also reduces the expression of transcription factors which are involved in the differentiation of adipocytes with a subsequent reduction in the accumulation of triglycerides, suggesting the role of mitochondria in the lipid metabolism of adipocytes at different levels [73]. The uncoupling of mitochondrial respiration can be performed by regulating the exchange of protons across the inner mitochondrial membranes, ultimately dissipating the gradient of protons to reduce the deleterious effects of ROS. The family of uncoupling proteins (UCP) present in the inner mitochondrial membrane play a significant role in the thermogenesis occurring in black adipose tissue (BAT), which helps in the regulation of mitochondrial ROS disposal in other tissues [74,75]. The UCP1 disengages mitochondrial respiration from the production of ATP by initiating proton leaks across the inner mitochondrial membrane, leading to the dissipation of energy in the form of heat. This process is heightened by NEFA and is impeded by purine nucleotides. The oxidative phosphorylation-mediated generation of ROS causes the activation of UCPs, which causes the dispersion of the proton gradient and leads to the removal of ROS [76]. Hence, this can affect the generation of ROS and can delay or even reverse their deleterious effects.

### 4.1. Contribution of Mitochondrial Dysfunction in Consumption of Calories and ROS

The oxidative dysfunction of mitochondria is correlated with the resistance developed towards insulin in skeletal muscles of diabetic and obese individuals and results in a decrease in size and number of mitochondria and their altered enzymatic oxidative capacity. A decrease in the expression of genes which mediate oxidative phosphorylation and the diminished consumption of oxygen is clearly witnessed in patients with obesity [77,78]. The adipocytes react to metabolic changes by altering the number, structure, and circulation of mitochondria within the cell, thereby modifying the content of metabolites, enzymes, and mitochondrial DNA [79]. An increase in the intake of calories poses a substantial load on mitochondria, which causes their dysfunction and leads to the effective dissipation of proton gradients, which further elevates the production of reactive oxygen species and produces cell damage, apoptosis, and mutations in mtDNA. The generation of ROS was found to increase upon the consumption of a high-fat diet and hyperglycemic conditions within adipocytes, leading to the abnormal production of adipokine. The consumption of oxygen in adipocytes is decreased by ROS, which further blocks the oxidation of fatty acids, resulting in the accumulation of lipids. The presence of mitochondrial antioxidants or the overexpression of scavengers of mitochondrial ROS have a tendency to mitigate insulin resistance [80,81]. Therefore, the presence of excessive energy substrates causes an elevation in the synthesis of ROS and poses a noteworthy outcome on the function of mitochondria and metabolism of its energy substrates. 

### 4.2. Role of Mitochondria in Brown and White Adipose Tissues

Mammals possess two major types of adipose tissue, namely brown adipose tissue (BAT), which releases energy in the form of heat, and white adipose tissue (WAT), which mainly specializes in the storage of energy. The adipocyte cells are obtained from multipotent mesenchymal stem cells, which reside in the stromal vascular fraction of adipose tissue [82]. Both WAT and BAT originate from distinct predecessor cells, and the differences in their functions in the metabolism of energy are also due to differences in the physiology of mitochondria. In scenarios where energy is demanded, WAT liberates NEFA as an energy substrate into the circulation, whereas during excessive nutrient conditions, the lipogenic enzymes that are present within WAT utilize the energy substrates to synthesize and store triglycerides [83]. The biogenesis of mitochondria and the expression of UCP1 proteins increases in WAT upon the stimulation of adrenergic pathways by exposure to cold. The number of brown adipocytes cells within the WAT varies distinctly and is mainly impacted by environmental factors. The adipocyte cells which are present in the BAT contain a Myf5-positive precursor protein, which is common with myocytes. The brown adipocytes, which are located in the WAT, are obtained from a distinct predecessor, namely Myf5 negative, which amplifies upon the stimulation of adrenergic receptors. The brown adipocytes evolve upon the differentiation of brown pre-adipocytes or via white adipocyte precursors or their further trans-differentiation. The BAT cells are thermogenic in nature and are dependent upon the stimulation of lipolysis via adrenergic receptors and NEFA degradation via UCP1 and aid in balancing energy in mammals. The muscular mitochondria and BAT possess similar metabolic profiles and have elevated oxidative capacity due to the increased density of mitochondria and the expression of FAO enzymes [84]. Overexpressed UCP1 proteins in the cells of WAT prevent weight gain by reducing the synthesis of fatty acids and increasing the expenditure of energy [85]. The BAT is located in several areas in humans and is activated upon exposure to cold and inhibited by the consumption of beta-adrenergic blocker drugs. The amount of BAT and its activity in humans is inversely proportional to age, levels of glucose, body mass index, and percentage of body fat. The cells within these tissues are the major contributors of thermogenesis in healthy adult humans. The predecessors of BAT have also been recovered in the skeletal muscles and further differentiate into mature brown adipocytes [86]. Hence, the BAT plays a crucial role in sensitivity towards obesity and regulates the expenditure of energy in humans and the processes that are linked to the functioning of mitochondria [87].

### 4.3. Transcription Factors in Adipocytes and Mitochondria

There is a growing interest in the exploration of the role of mitochondria in adipocyte differentiation and their implications in the therapeutic management of obesity. There are several transcription factors which participate in the process of adipogenesis, such as peroxisome proliferator activated receptor (PPAR), gamma coactivator (PGC) family, PRD1-BF-1-R1Z1 homologous domain comprising protein 16 (PRDM16), adipokines, and growth factors. Transcriptional coactivators such as PGC-1a and PGC-1b play a significant role in the expression of genes that are involved in the biogenesis of mitochondria, metabolism of fatty acids, and accumulation and aggregation of lipids [88]. A decrease in the levels of PGC-1 impairs the functioning of mitochondria and increases the accumulation of lipids, which is a major characteristic of this metabolic disorder. PRDM16 is a transcriptional coactivator of PGC-1α and PGC-1β, which enhances the genetic expression of proteins that are involved in the biogenesis of mitochondria, induces their uncoupling and oxidative phosphorylation, and is exclusively found in brown adipocytes. The overexpression of PRDM16 in adipocytes transgenically elevates the genetic expression of mitochondria in BAT-like cells within the WAT and hence is stated to be a primary moderator of brown adipogenesis by increasing the biogenesis of mitochondria, uncoupling, and energy expenditure. The human WAT plays a prominent role in the endocrine system as it aids in the production of hormones and adipokines which regulate energy homeostasis and affect the functioning of mitochondria [89,90]. Adiponectin majorly affects the metabolism of lipids and glucose, alters insulin sensitivity, and stimulates FAO and the uptake of glucose into skeletal muscle cells. 

## 5. Pathophysiological Role of Mitochondrial Dysfunction in Infectious Diseases

The mitochondrial system plays a significant role in several physiological processes of the human body. Loss of mitochondrial integrity leads to the stimulation of diverse signaling pathways which control the cellular response to external infectious antigens. The stimulation of search mitochondria-related cellular responses has implicated its role in controlling signaling and recognition of the pathogen-mediated onset of infection. Inflammation in the cells of the human body occurs as a complex response towards infection and represents the extent of damage [91]. The occurrence of infection acts as a stimulus for morphological changes that occur within mitochondria. For instance, upon infection with *Vibrio cholerae*, the fragmentation of mitochondria is triggered due to the secretion of VopE factor. Bacterial infection with *Helicobacter pylori* releases vacuolating cytotoxin A, which stimulates the Drp1-mediated fragmentation of mitochondria [92]. The mechanism of mitochondrial fission has been used by the above infection toxins, which causes the fragmentation of the mitochondrial network and leads to cell death. The bacterial infections not only destroy the network of mitochondria but also induce the elongation of the organelle, as presented in *Chlamydia trachomatis* infection, which downregulates the levels of DRP1 protein by inducing host cell micro-RNA expression. The apoptotic pathway of mitochondria displays its role in the signaling of cellular responses towards cytosolic stress [93]. The antiviral response of mitochondria in response to the activation of the mitochondrial membrane protein MAVS also presents the ability of the organelle to transmit signals and downstream the release of proinflammatory cytokines and interferons. 

### 5.1. Infection Dynamics and Structure of Mitochondria

An exclusive role of mitochondria in cell-autonomous immunity has been established. Bacterial infections have shown fragmentation and fusion of mitochondria, whereas in viral infections, hyperfusion and fragmentation have been described. Upon infection with severe acute respiratory syndrome-related coronavirus (SARS-CoV2), or human immunovirus (HIV), an increase in mitochondrial fusion has been reported, whereas Hepatitis B and C and the influenza virus have shown mitochondrial fission. The fragmentation of mitochondria has also been observed upon exposure to parasitic infections, such as with *Toxoplasma gondii*. The morphological differences in the network of mitochondria affect the generation of energy and can serve as an additional marker for different types of cells [94,95]. Dendritic cells and inflammatory macrophages contain elongated mitochondria, whereas proinflammatory macrophages or B cells contain a fragmented mitochondria network. The role of mitochondria in relation to infectious diseases is highly complex and is being studied. The influenza virus has been reported to initiate the fragmentation of mitochondria alongside the elongation of mitochondria by inhibiting the recruitment of Drp1 protein. The pharmacological initiation of the fragmentation of mitochondria has been found to reduce the viral burden, demonstrating the significance of the elongation of mitochondria in the replication of a virus. Listeriolysin O, a pore-forming toxin found intracellularly in *Listeria monocytogenes*, causes mitochondrial fragmentation and elevates the levels of MICOS component Mic10 in secluded mitochondria. The fragmentation of mitochondria is inhibited upon a deficiency in Mic10 due to *Listeria* infection, whereas its overexpression causes the fragmentation of mitochondria independent of the status of infection [96].

### 5.2. Metabolism of Mitochondria during Infection

Cellular energy is gained through the fermentation of lactic acid, glycolysis, and the respiration of mitochondria. The stimulation of lipopolysaccharides or other associated macrophages during infectious conditions causes metabolic transformation of macrophages by regulating the TCA cycle and glycolysis, which is mainly linked to the proinflammatory and anti-inflammatory phenotypes of the macrophagic molecules. These cellular and mitochondrial metabolic changes greatly impact the signaling pathways and present significant effects on infections associated with individual pathogens [97]. The cells that depend upon the respiration of mitochondria for the production of energy possess a limited ability to recycle plasma membrane proteins and serve as receptors for pathogens. Under high mitochondrial respiration conditions, the receptors gather intracellularly, causing an alleviated level of receptors on the surface, reducing the invasion of viruses and bacteria due to a lesser count of receptors [98]. Therefore, it can be concluded that the direct impact of infection is linked to the energy produced due to the metabolism of mitochondria. Zika virus infects astrocytes and the cells of the cerebral cortex and further replicates within the endoplasmic reticulum, causing an imbalance in the production of reactive oxygen species, the defective respiration of mitochondria, and an altered synthesis of ATP. The release of calcium upon infection with Hepatitis C virus from the endoplasmic reticulum also represents a signal for excessive ROS production by mitochondria, indicating the alterations in the metabolism of mitochondria in response to infections. The altered metabolism of mitochondria also affects inflammation in a non-cell-autonomous manner [99]. Considering a direct relation between the metabolism of mitochondria and the secretion of inflammatory cytokines, the response to infection is probable to induce inflammation and immunity in the affected cells.

The role of mitochondria in apoptosis has been well established. Bcl-2 proteins are hosted by mitochondria in their external membrane, and further release intermembrane space protein cytochrome c. In the cytosol, these proteins trigger caspases and cause apoptosis. The association between mitochondria and pathogens is multifaceted. The dysfunction in mitochondria alters immune processes in response to pathogens [100]. The various roles of mitochondria in the cells of mammals have been established and are currently a topic of great interest due to their complex role in activating several autonomous signaling pathways. Hence, understanding their significance in regulating immunity against infectious diseases could further lead to the development of new treatments for the management of infectious diseases.

## 6. Conclusions

Mitochondria are the main source of energy for the functioning of the cell and the human body and are known as the metabolic hub of the cell. Impairment in this crucial organelle can be witnessed at a broader systemic level. Several studies have established that mitochondria, apart from generating power for the efficient functioning of the body, are also involved in several other additional processes in the body, thereby posing a huge influence on the functioning and dynamics of the various ongoing processes in the human body. Therefore, advanced therapeutic mechanisms based on the physiological role of mitochondria are being developed by identifying various predecessors of metabolic diseases which can be targeted for the management of the diseases. The above review highlighted the principal role of mitochondria in the pathological processes of the development of chronic diseases, visualizing a new treatment approach for their management.

## Figures and Tables

**Figure 1 jcm-12-02882-f001:**
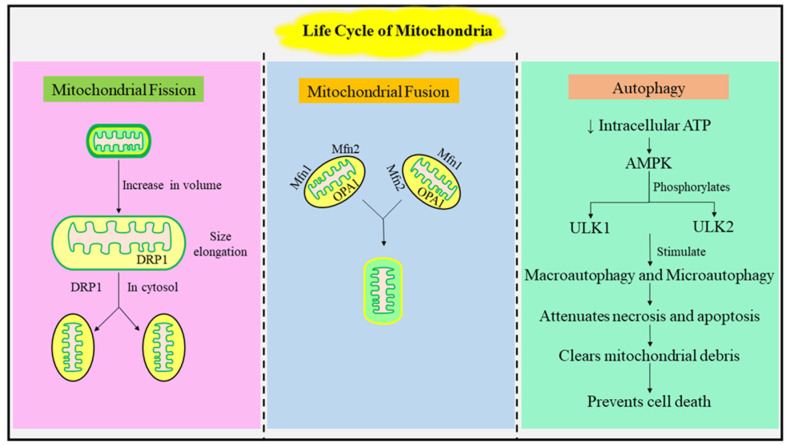
The above figure briefly represents the stages in the life cycle of mitochondria. The figure begins from the left by explaining the process of mitochondrial fission, which involves the division of a mitochondrion into two separate mitochondrial cells, followed by mitochondrial fusion in the middle which involves the fusion of two separate mitochondria into one. On the right, autophagy, which regulates the functioning of the mitochondria, is shown.

**Figure 2 jcm-12-02882-f002:**
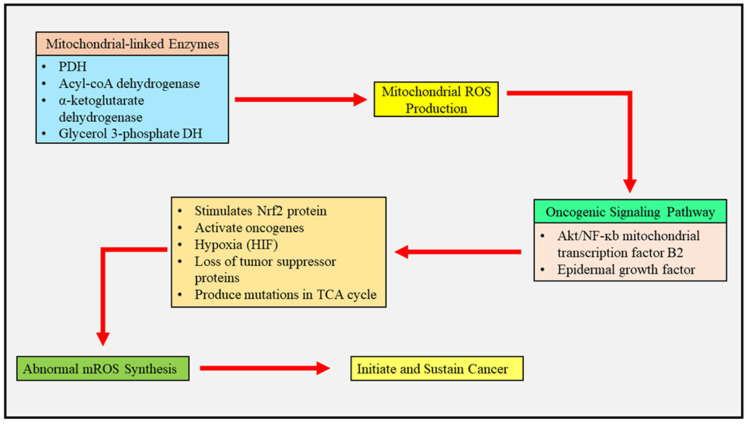
The above figure highlights the role of reactive oxygen species in cancer. Several mitochondria-linked enzymes, such as pyruvate dehydrogenase (PDH), acyl-coA dehydrogenase, α-ketoglutarate dehydrogenase, and glycerol-3-phosphate dehydrogenase, initiate the production of mitochondrial reactive oxygen species, which induce the signaling of oncogenic pathways and lead to the abnormal synthesis of mitochondria, which further initiates and sustains cancer.

**Figure 3 jcm-12-02882-f003:**
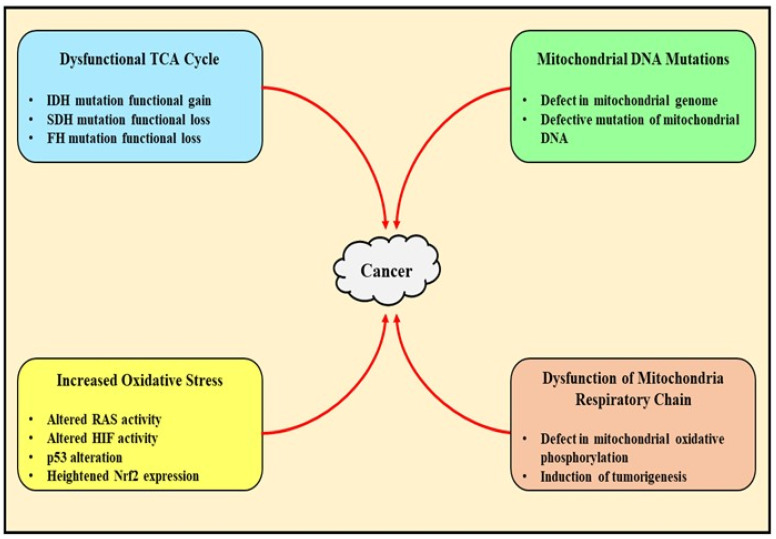
The above figure highlights the different factors involved in the pathophysiology of cancer, such as the dysfunction of the TCA cycle, increased oxidative stress, and mutations and dysfunction in mitochondrial DNA and the mitochondrial respiratory chain.

**Figure 4 jcm-12-02882-f004:**
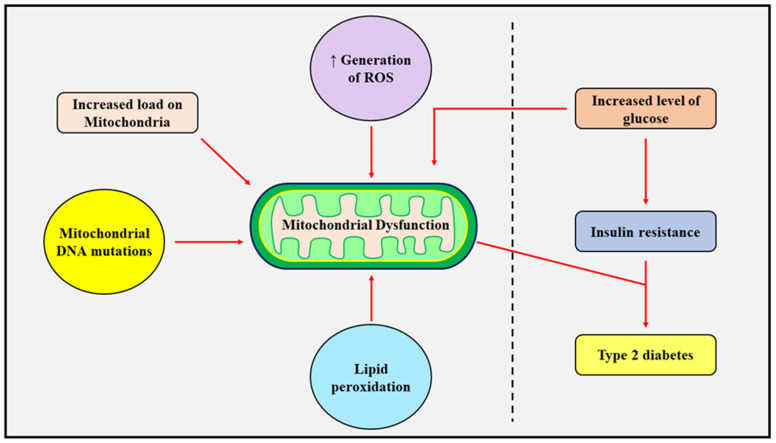
The above figure highlights the factors involved in the initiation of diabetes, which includes the increased production of reactive oxygen species, mutations in mitochondrial DNA, and the peroxidation of lipids, which increases the load on mitochondria and causes an increase in glucose levels and insulin resistance.

## Data Availability

Not applicable.

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
