# Peer review of "Mitochondrial Dysfunction: A Cellular and Molecular Hub in Pathology of Metabolic Diseases and Infection"

_jcm, 2023, doi:10.3390/jcm12082882_

Round 1

Reviewer 1 Report

The English writing which is beyond the scope of a reviewer needs to be improved. The review needs to be improved with more insights and critical discussion. There are many brief sentences not supported by literature.

The review only included a few recent literatures. Studies carried out in recent years should be added and a detailed discussion should be presented.  Figures on the subject should be increased.

The first 3 paragraphs of part 3 are too long. subheadings should be added to make them clearer and readable.

Inappropriate use of abbreviations. All nonstandard abbreviations should be defined at first use in the text.

Author Response

Reviewer 1

The English writing which is beyond the scope of a reviewer needs to be improved. The review needs to be improved with more insights and critical discussion. There are many brief sentences not supported by literature.

As stated by the esteemed reviewers, English writing has been improved, with more improved insights and critical discussion supported by literature and highlighted in green.

The review only included a few recent literatures. Studies carried out in recent years should be added and a detailed discussion should be presented.  Figures on the subject should be increased.

As stated by the esteemed reviewers, literature citations have been updated and figures have been added and highlighted in green.

The first 3 paragraphs of part 3 are too long. subheadings should be added to make them clearer and readable.

As stated by the esteemed reviewers, subheadings have been added in first 3 paragraphs of part 3 and highlighted in green.

Inappropriate use of abbreviations. All nonstandard abbreviations should be defined at first use in the text.

As stated by the esteemed reviewers, all nonstandard abbreviations should be defined at first use and highlighted in green.

Reviewer 2 Report

Authors wrote in the abstract :

The dysfunction of mitochondria due to mutations in mtDNA can malfunction the TCA cycle and cause leakage of electron respiratory chain leading to increase in reactive oxygen species and signaling of aberrant oncogenic and tumor suppressor proteins which further alter the pathways involved in metabolism, disrupted redox balance and induce endurance towards apoptosis and several treatments which play a major contribution in developing several chronic metabolic conditions.

Sentence need to be adjusted as nuclear DNA variants affecting genes which are coding for mitochondria proteins may have the same impact.

Fusion fission and autophagy section

This section is clear but I invite the authors to describe in detail the autophagy mechanisms.

Figures will be welcome.

Metabolism of Mitochondria Within Different Tissues

This section is very incomplete, more details are needed. Otherwise this one will have to be removed.

Mitochondrial Reactive Oxygen Species and Cancer Metabolism

The section is well written. However, figure will be useful. The authors need to explain the inverse correlation between ROS production and ATP synthesis.

The authors focus on cancer diabetis and obesity. We can expect more example such as infectious disease.

A section on functional validation of mitochondria variants is lacking.

The authors need to investigate the cooccurrence between mitochondria variants and nuclear variants.

Author Response

Dear reviewer

Please find the attached word document for the resolved comments.

Best regards

Round 2

Reviewer 1 Report

This review summarizes the roles of mitochondrial dysfunction in cancer, diabetes mellitus, infections and obesity. However, there are several concerns that need to be addressed:

1: The discussion of mitochondrial dysfunction in each aspect is too general, and there are already many similar reviews published. What is the novelty and value of this manuscript? The authors need to provide a clear justification for why this review is necessary and what new insights it brings to the field.

2: The majority of the literature cited in the review is quite old, with only approximately 10% published within the past five years. This raises doubts about the up-to-dateness and relevance of the information presented. To ensure that the review is current and credible, at least 50% of the cited literature should be within the past five years.

3: The manuscript needs improvement in terms of language quality. The English used is not authentic and some long sentences are confusing. The authors need to check the grammar, punctuation, and clarity of their writing.

Author Response

This review summarizes the roles of mitochondrial dysfunction in cancer, diabetes mellitus, infections and obesity. However, there are several concerns that need to be addressed:

1: The discussion of mitochondrial dysfunction in each aspect is too general, and there are already many similar reviews published. What is the novelty and value of this manuscript? The authors need to provide a clear justification for why this review is necessary and what new insights it brings to the field.

As stated by the esteemed reviewers, a brief novelty of the manuscript has been added and highlighted in blue.

The current article presents a recent comprehensive review of the role of mitochondrial dysfunction in onset of several chronic life altering disorders such as cancer, diabetes mellitus, obesity and infections. The article explains in brief the life cycle of mitochondria including its processes of fusion, fission and autophagy. The article also highlights the role of mitochondria in metabolic processes of the body, and concisely describes its validation and cooccurrence with nuclear DNA. Extensive study of the recent literature was performed for preparation of the article to present the readers with the various roles of mitochondria in management of several chronic disorders elaborately discussed in a single review paper which brings novelty to the article.

2: The majority of the literature cited in the review is quite old, with only approximately 10% published within the past five years. This raises doubts about the up-to-dateness and relevance of the information presented. To ensure that the review is current and credible, at least 50% of the cited literature should be within the past five years.

As stated by esteemed reviewers, over more than 50% of the cited literature has been revised and updated as within the past five years.

3: The manuscript needs improvement in terms of language quality. The English used is not authentic and some long sentences are confusing. The authors need to check the grammar, punctuation, and clarity of their writing.

As stated by the esteemed reviewers, language of the manuscript has been improved and changes have been highlighted in blue.

Reviewer 2 Report

Manuscript has been improved

I endorsed it

Author Response

The author thanks the reviewers for their esteemed review that led to betterment of the manuscript.